# Not Too Warm, Not Too Cold: Thermal Treatments to Slightly Warmer or Colder Conditions from Mother’s Origin Can Enhance Performance of Montane Butterfly Larvae

**DOI:** 10.3390/biology11060915

**Published:** 2022-06-15

**Authors:** Konstantina Zografou, George C. Adamidis, Brent J. Sewall, Andrea Grill

**Affiliations:** 1Institute of Ecology and Evolution, University of Bern, Baltzerstrasse 6, CH-3012 Bern, Switzerland; adamidis@upatras.gr (G.C.A.); a.grill@univie.ac.at (A.G.); 2Laboratory of Plant Physiology, Department of Biology, University of Patras, 26504 Patras, Greece; 3Department of Biology, Temple University, Philadelphia, PA 19122, USA; bjsewall@temple.edu; 4Department of Evolutionary Biology, University of Vienna, Djerassiplatz 1, A-1030 Vienna, Austria

**Keywords:** butterflies, mountain species, invertebrates, climate change, fitness, alpine

## Abstract

**Simple Summary:**

Extreme weather events and climate change can alter organismal development and, in turn, affect species survival, community composition, and ecosystem processes and services. We examined the performance of butterfly larvae of five montane *Erebia* species from the Swiss Alps under three thermal scenarios: at, above, or below those at the elevation where their mother originated. We found evidence of better larval performance in temperature treatments associated with low and middle elevations and a decreased performance at temperature treatments associated with higher elevations. In contrast, larvae performed poorly in thermal treatments that differed strongly from maternal conditions. The inclusion of additional life history stages in future studies could further advance the understanding of factors affecting thermal tolerance in cold-adapted *Erebia* butterflies.

**Abstract:**

Climate change alters organismal performance via shifts in temperature. However, we know little about the relative fitness impacts of climate variability and how cold-adapted ectotherms mediate these effects. Here, we advance the field of climate change biology by directly testing for species performance, considering the effects of different thermal environments at the first developmental stage of larvae. We conducted our experiments in climatic chambers (2019–2020) using five cold-adapted butterflies of the genus *Erebia* (*Erebia aethiops, Erebia cassioides, Erebia manto, Erebia tyndarus, Erebia nivalis*). Larvae were reared indoors and were treated with higher and lower temperatures than those of their mothers’ origins. Overall, we found evidence of better performance at warmer temperatures and a decreased performance at lower temperatures, and larvae were able to tolerate small temperature changes from mother’s origin. Warmer conditions, however, were unfavorable for *E. nivalis*, indicative of its limited elevational range and its poor ability to mediate a variety of thermal conditions. Further, larvae generally performed poorly where there was a large difference in thermal regimen from that of their maternal origin. Future efforts should include additional life history stages and focus on a more mechanistic understanding of species thermal tolerance. Such studies could increase the realism of predicted responses to climate change and could account for asynchronous changes in species development, which will alter community composition and ecosystem functioning.

## 1. Introduction

Temperature is known to affect many aspects of physiology, growth, performance, and fitness in ectothermic organisms [1]. Particularly in alpine regions, the effects of a warming climate may be both acute and complex [2,3]. For instance, air temperature in the European Alps has increased at a rate of about 0.36 °C decade^−1^ since 1970, leading to glacial retreat and significant snowpack reduction [4]. As a result, this mountainous region is undergoing marked changes in spring phenology and in the elevational distribution of animals and plants [4]. Such rapid and diverse shifts will alter local microclimatic conditions, food availability, and species interactions, posing numerous challenges to ectotherms’ performance and survival.

Ectotherms’ responses to such climate-driven environmental changes vary and include rapid upslope and downslope movements [5], relocation to different habitat types [6], and phenological variation [7,8]. Upslope movements are common: for example, 56% of the population of the butterfly *Erebia cassioides* occurred above the treeline in 1975, while 29 and 37 years later, 99% of the population was above the treeline [9]. However, downslope movements are also possible [10], such as when alpine species move to lower elevations, as newly formed forest openings become more available there [11]. Microclimatic variation also affects organismal response to warming at a larger scale: Suggit et al. [6] showed that in hot years, butterflies shift into cooler, closed habitats such as woodlands, and in cooler years, they shelter in warmer, open habitats such as grasslands. Furthermore, diapause termination of alpine caterpillars is often influenced by snowmelt, and the advanced flight of the alpine butterfly, *Erebia epiphron*, was attributable to shorter winters or to earlier snowmelts [12]. Shortened periods of snow cover may even cause mortality, as the buffer layer of snow under which insect larvae spend the winter in diapause decreases [13].

In Lepidoptera, the larvae, especially during the early immature stages (egg, larva, pupa), have limited or zero mobility [14] and may be unable to move to alternate habitats. Hence, the mother’s oviposition locations determine the conditions that the offspring will experience, and these consequently influence larval development and survival [15,16]. As further climatic fluctuations and shifts in the elevational limits of species’ ranges are anticipated [17,18], the study of organisms at critical early life stages under different climatic conditions are required to clarify the relationships among temperature shifts and species’ thermal tolerances and performance and capacity for evolutionary adaptation. Such studies may have important implications, since lepidopterans account for a large proportion of earth’s biodiversity [19], play a key role in energy flow within ecosystems [20,21], and are the dominant leaf herbivores of many forest ecosystems [22]. 

The current distribution of ectotherms is often assumed to define the upper and lower limits of their thermal tolerance, and thus is used to predict future response to climate change [23]. However, thermal limits may differ from the thermal range that species currently occupy in ways that reflect ecological variation or past evolutionary history. For instance, ectotherms in the tropics have a lower thermal tolerance than temperate species, due to the narrower thermal safety margins that tropical ectotherms display [24]. A recent study revealed that ectotherms originating from cold paleoclimates have lower cold tolerance limits than those with warm thermal ancestry [25]. Accordingly, Vrba et al. [5] found a reversed altitudinal cline in cold hardiness between an alpine (*E. tyndarus*) and a low-elevation butterfly (*E. medusa*). Thus, current thermal ranges inhabited by a species are far from perfect predictors of a species’ ability to respond to climate shifts, and a better understanding of species’ thermal limits is needed to predict more accurately species’ responses to climate change.

This study aims to improve our understanding in the field of climate change biology by directly testing for species performance, taking a comparative approach between five species, and considering the effects of temperature treatments at immature life stages. In particular, we sought to explore whether changes in temperature can alter the performance of larvae among species in the butterfly genus *Erebia* Dalman, 1816. We hypothesized that movements from mother’s origin to lower elevations (represented in our study by higher temperatures) will positively affect species performance, as increasing temperatures can lead to higher metabolic rates and quicker development [26,27]. In contrast, movements to higher elevations (represented in our study by lower temperatures) will supposedly slow species performance. Furthermore, we hypothesized that strictly alpine species will display a low thermal tolerance across different temperature treatments, resulting from the buffering function of snow cover [28]. In contrast, species with a broader elevational distribution are expected to demonstrate a greater thermal tolerance and a better performance across the different temperature treatments due to their adaptation to more irregular and unpredictable snow cover in lower elevations.

## 2. Materials and Methods

### 2.1. Study System

Our study area was in the Bernese Alps of Switzerland. The Bernese Alps are a subset of the 1200 km-wide mountain range of the Alps, located between the Swiss cantons of Bern and Valais. With an altitudinal range between 2000 and 4270 m and a variety of mountainous habitats such as rocky corridors, grasslands and open woodlands, this area is optimal for the study of alpine butterflies of the genus *Erebia* [29]. The species-rich genus *Erebia* (Lepidoptera: Nymphalidae, Satyrinae) includes 100 described species, 26 of which occur in Switzerland [30], and it is thus the most diverse butterfly genus in the Palaearctic [31]. *Erebia* species inhabit northern alpine cold environments, where rocky grasslands and open woodlands predominate [32]. 

*Erebia* butterflies (Ringlets) are univoltine or semivoltine species with adults flying from May to September, depending on species and elevational range. They are good indicators of climate change [33], and several representatives occur sympatrically, providing a good system for studying thermal adaptations in changing environmental conditions [28,34,35]. The larvae feed on grasses or sedges, and in many alpine species, the development takes two or three years [29]. Several hypotheses for the prolongation of larval development in alpine butterflies refer to a combination of both biotic (short vegetation period and hence short duration of food intake), and abiotic (unfavorable thermal conditions) factors that characterize alpine ecosystems. Others suggest the cost of parasitoid avoidance [36] or reflect various postglacial histories [37], but further investigation is needed the light of climate warming. 

### 2.2. Temperature Treatments

Long-term climatic data (1993–2019) were obtained from the Federal Office of Meteorology and Climatology of Switzerland (MeteoSwiss). We selected eleven meteorological stations that covered the whole study area, and we corresponded averaged values of air temperature to three elevational zones (low: 1655–1987 m, medium: 1988–2320 m, high: 2321–2655 m). Elevational zones resulted after dividing the elevation range (1655–2653 m) of the sampling system into three equal segments. Every week, we changed the thermal treatments (in the three chambers; see below) to match the weekly day/night temperature for the elevational zone, averaged from the long-term hourly dataset, and starting on the third week of June (Appendix A). In addition, the day–night rhythm of the illumination was adjusted weekly according to the averaged weekly day length for 2019 (week 1–2, 15.5 h; week 3, 15.4 h; week 4, 15.3, week 5, 15.2 h; week 6, 15 h; week 7, 14.4 h; week 8, 14.2 h) (Appendix A). A gradual transition of one hour was applied between day and night conditions (and vice versa). For analytical purposes, however, averaged values were finally used as provided in Appendix A. Caterpillars were weighed in a Mettler Toledo XS105 DualRange scale (±0.01 mg), and every other day, we changed the paper filter that was placed at the bottom of the petri dishes to clean any presence of feces or food that could influence the caterpillars’ artificial environment.

### 2.3. Experimental Design

We started our sampling at 1655 m, and moving upward toward higher elevations (until 2653 m), we collected females of all *Erebia* species found flying between 19 August and 4 September 2019. We sampled 10–15 females from each species at 10 localities (lat. 46.3745–46.767503 N, long. 7.754379–9.620249 E) in the study area. After each visit, individuals were transferred to climatic chambers with optimal conditions of 20/10 °C and 16/8 h of light to increase activity and speed up oviposition. Only the ones that survived and gave offspring were selected for the study, limiting our pool to five species.

*Erebia aethiops* represents a lower-elevation species that is distributed between 340 and 2100 m [38]. It flies from July until September and can be found in a wide variety of habitats, such as dry meadows, pastures with tall grass above the timberline and open woodlands [39]. It is polyphagous with one generation per year and hibernates in the first or second instar (L1/L2) of the development stage [29]. *Erebia cassioides* represents an alpine species that is distributed between 1600 and 2600 m and inhabits grassy and rocky slopes around and above the timberline. Its larvae overwinter in the first or second instar and complete a life cycle within a year [29]. *Erebia manto* is a typical alpine species, restricted to the mountain ranges of the western Palaearctic region. It hibernates two times per life cycle—the first time as an egg or in L1 developmental stage and the second in L4 stage. It is a rather common species, occurring in most of the European mountains, and specifically in pastures above the timberline zone between 1200 and 2540 m [29]. *Erebia tyndarus* is a univoltine species that overwinters in the L1/L2 developmental stage. It lives in dry and sunny places in the central parts of the Alps and ranging between 1200 and 2800 m [29]. Lastly, *Erebia nivalis* is a strictly alpine species, occurring over the timberline zone in sunny and dry places that range between 2260 and 2800 m. It hibernates twice per life cycle—the first time is in the L1 developmental stage and the second in the L3 stage [29].

Each female was allowed to lay eggs in a 1-L plastic, transparent container with a net lid. The net served as barrier to contain the butterfly and was used as the main substrate for oviposition. The butterflies were fed through a paper soaked with orange juice, which was renewed daily. In addition, the nets and the rest of the containers were meticulously checked for eggs every day. Oviposition lasted 39 days (from 23 August until 1 October) during which a total of 1129 eggs were collected. Newly hatched caterpillars in the L1/L2 developmental stage were transferred to individual petri dishes before being transferred to hibernation rooms. We placed them into hibernation to better simulate their life cycle: our species, similar to most *Erebia* species, overwinter as larvae and hibernate in grass tussocks [29]. Therefore, hibernation started on October 15th, 2019 and lasted 28 days, where specimens were held at −1 °C in dark conditions. Body mass was measured before and after hibernation, and weekly for the following eight weeks. After our last measurements on body mass where most individuals had reached the last stage (L4), we stopped monitoring larval growth. After hibernation, caterpillars from each species and location were equally divided and left to grow in three different chambers, where thermal conditions simulated the conditions of mother’s origin corresponding to three elevation zones (see below for an example). In this sense, if a mother was originally collected in the lowest elevation zone, her offspring were left to grow in temperatures associated with the lowest elevational zone (mothers’ origin), but it was also moved in temperatures associated with the medium (up1) and highest (up2) elevation zones. Accordingly, if a mother was originally collected in the medium elevation zone (mother’s origin), her offspring were also moved to grow in the highest (up1) and lowest (down1) zones (see Appendix A). In nature, *Erebia* caterpillars are known to emerge from winter diapause in June [29]. Considering the elevation range of our study, we attempted to maintain the seasonal conditions by first taking the mean temperatures of the third week of June to set the temperatures for the emerging point of our caterpillars.

#### Diet

Because alterations in thermal environments are known to affect nutritional quantity and/or quality (e.g., host plant nutrient concertation) which in turn can greatly contribute to variation in performance measurements such as body mass [40], we standardized the diet using lab-produced host plants [41,42,43]. 

At the beginning of July 2019, we started growing a grass plant, *Festuca pratensis*, one of the host plants for our studied organisms, in normal potting soil (substrate 144, Ricoter, Aarberg, Switzerland) in climate chambers mimicking the temperature conditions of the three elevational zones as described above. We used square pots with a soil surface of 144 cm^2^, and we sowed 0.5 g of *F. pratensis* seeds (UFA-Samen, Winterthur, Switzerland) in each one of them. The pots were regularly watered in intervals of three to four days. Every other day, two leaves were placed in the dish and cut into smaller pieces to occupy as much area as possible. Because caterpillars are sensitive to food quality [44], we provided only the healthy leaves and avoided the dry or damaged ones. The caterpillars were placed over the leaves or surrounded by them, and every day, the filter paper was moisturized with tap water.

### 2.4. Statistical Analysis

To gain a better insight on species performance and thermal tolerance, we modeled their performance, the phenotypic trait of body mass expressed by individual organisms inhabiting mountainous and alpine ecosystems, as a function of three different thermal conditions while keeping constant the quantity and quality of the food source. Specifically, performance was modeled as a function of time, species, temperature treatments, upslope/downslope movements, mother’s elevation and two-way interactions of species by time, temperature treatment by time, mother’s elevation by temperature treatment, and time by upslope/downslope movements. Since the measures of larval mass in subsequent time intervals are not independent, in other words, a relatively large individual in a given time step would likely still be relatively large a week later, we considered the AR-1 auto-correlation structure in our model [45]. We fitted a linear mixed effects model, with “individual” as a random effect to avoid pseudoreplication, with lme4 package v 1.1-26 [46] in R version 4.1.2 [47]. Since hatching time slightly varied among larvae, we considered the initial post-hibernation larval mass to our weekly measurements. For model selection, we used Akaike’s Information Criterion (AIC) starting with the more complex terms (i.e., interactions) and moving on to single term effects [45]. A natural log transformation was applied to *Erebia* larval mass such that the assumptions of normality were met. To account for differences in maternal quality, individuals were nested to mothers, but this model structure increased AIC and was discarded. In addition, we analyzed larval mass (or biomass hereafter) as a function of temperature treatments and time for each species separately. After fitting our models, we conducted post hoc comparisons for all combinations of a single factor or an interaction of a covariate and a factor in *emmeans* package [48]. The package provides estimations of marginal means (predicted values) for the response at the margin of specific values or levels from certain model terms. For graphical display and single factors, we used *ggeffects* package [49] and *emmip* function, while figures for continuous variables were plotted in ggplot2 [50]. Finally, for the statistical analysis, we excluded data from individuals that did not survive the first week after hibernation (29 records were discarded).

## 3. Results

Our model fitted five *Erebia* species and 199 individuals (see Appendix A) and highlighted the importance of time and upslope/downslope movements interaction (χ^2^ = 11.69, df = 4, *p* = 0.02), time and thermal treatment interaction (χ^2^ = 9.19, df = 2, *p* = 0.01) and the effect of the mother’s elevation (χ^2^ = 14.01, df = 1, *p* < 0.001) (Appendix A). Comparisons revealed that, on average, high and medium temperature treatments improved species performance but not for the low temperature treatments (Figure 1). Furthermore, post hoc comparisons revealed a difference on species performance between movements of opposed directions (e.g., down1–up1, down1–up2, down2–up1, down2–up2) as well as movements of two-step differences (e.g., no change–up2) (Figure 2). The only exception was the difference we found on one-step movements toward the same direction (up1–up2) (Figure 2). Overall biomass was also found to decrease along mother’s elevation axis (mean ± 1 SE = −1.08 ± 0.003, Figure 3).

Individual tests showed that *E. aethiops* performed better at mid and high temperature treatments with time (χ^2^ = 6.64, df = 2, *p* = 0.04, Figure 4A, Appendix A) and *E. cassioides* performed better at mid temperature treatments with time (χ^2^ = 30.54, df = 2, *p* < 0.001, Figure 4B, Appendix A), while *E. tyndarus* showed higher biomass, on average, at medium treatment temperature (χ^2^ = 9.28, df = 2, *p* = 0.01, Figure 4C). In addition, *E. aethiops*’ performance was marked to be negative on the low temperature treatments (Figure 4A, Appendix A). There was no difference in the performance of *E. manto* among temperature treatments (χ^2^ = 2.54, df = 2, *p* = 0.28), only an increased biomass across time (χ^2^ = 4.3, df = 1, *p* = 0.04). Lastly, *E. nivalis* showed a decreased biomass in higher thermal conditions (χ^2^ = 14.54, df = 2, *p* < 0.001) when compared to that of lower or mid temperature treatments (Figure 4, Appendix A).

## 4. Discussion

By examining the performance of five *Erebia* species under different thermal conditions, we tested several underlying hypotheses about the potential performance of the first developmental stage under climate change. We found evidence of a better performance at temperatures associated with mid and low elevation treatments and a decreased performance at temperatures associated with higher elevation treatments. An interesting finding was that large movements to either warmer or colder conditions can seriously hinder species performance. Finally, we underlined that species-specific analyses must be applied to adequately capture the character of this cold-adapted taxon.

We found *Erebia* butterflies to outperform in low and mid-elevation thermal conditions. A good performance under warmer conditions was expected, as high temperatures are well-known to boost metabolism and enhance development [27]. However, when this cold-adapted group was treated under lower temperatures, it decreased its performance, suggesting a lower thermal tolerance under these conditions. A recent study found that ectotherms originating from cold paleoclimates have lower cold tolerance limits than those with warm thermal ancestry [25]. This suggests that even cold-adapted ectotherms may be able to respond better to heat stress than cold stress [25].

There are observations of species moving toward higher elevations in response to rising temperatures [51,52,53,54] but also species colonizing lower elevation biotopes [28]. To this extent, if a species lives in a region below its thermal optimum, some warming can increase its performance and vice versa [55]. The ability of studied organisms to tolerate one-step movement to slightly colder or warmer locations is indicative of a higher performance. If, however, movements are larger—stepping further down or above the mother’s origin (such as the two-step movement in our study)—performance can be reduced. Here, what seems to drive the decreasing performance is mainly the upslope/downslope movements of low-elevation species (*E. aethiops*) and strictly alpine species (*E. nivalis*), respectively.

Further analysis on *E. nivalis* showed that its biomass drops when individuals remain growing for eight weeks in the highest thermal conditions, more than when *E. aethiops* was moved to the lowest temperatures, stressing the importance of elevational distribution when thermal adaptability is questioned. Since this strictly alpine species is preadapted to colder conditions, it was not unexpected that its developmental process will not be enhanced by warmer conditions. Furthermore, increased energy use over the warmer conditions could negatively influence adult fitness, assuming that this energy cost counteracts any accelerated development associated with higher metabolic rates during the higher temperatures we provided [56].

Individual biomass largely increased with time for all, except from the strictly alpine *E. nivalis*. As *E. nivalis* reaches maturity in two years or even three years [29], it might be possible that the first developmental stage (L1) feeds little before entering diapause. Our experimental design enabled one hibernation period, and the lack of time effect compared to species that reach imago after one hibernation (e.g., *E. aethiops*) might be indicative of a lower metabolic rate and a restricted need to feed and speed up development [57]. Our analysis on *E. manto*, however, showed that temperature treatments had no effect on larvae’s mass. This difference between the two semivoltine species (*E. nivalis* and *E. manto*), may be attributed to differences in historic temperature exposure, resulting from different distribution patterns. It is possible that the wider elevational distribution of *E. manto* (1200–2540 m) provides a greater adaptability to larvae, ensuring the maintenance of high performance across a greater range of temperatures.

In the chamber experiments, performance of *E. tyndarus*, *E. cassioides*, and *E. aethiops* caterpillars was enhanced by the warming to which they were exposed. Individuals of these species were 20% larger in warmer conditions of mid and/or high temperature treatments than in low temperature conditions. As thermal performance curves are asymmetrical, organismal performance can be affected more drastically by the shifts in high rather than low temperatures [58]. Therefore, an interesting question would be whether the faster growth under higher temperatures can compensate for the extra cost. Growing faster during the first developmental stages may be nullified by higher metabolic costs during winter when larvae hibernate. For example, an experimental study found that butterfly larvae used 43% less energy in cold than in warm conditions over winter, suggesting that there is a tradeoff in overwinter metabolism resulting from local adaptation to cooler conditions [59].

We found total biomass to decrease with mother’s elevation. Despite some drawbacks in our experiment (last larval stage, pupa and imago’s performance were not monitored), the higher caterpillars’ size when reared in warmer conditions might be indicative of a reversed temperature-size rule [60], as the sizes of larvae and adults are likely to be correlated [22]. In the context of shrinking body sizes in ectotherms [61] and the higher impact of global warming on mountain ecosystems [62], ideally, more mechanistic studies, such as this one, need to be conducted on different life history stages to glean a better understanding of factors controlling for mature size shifts. The inclusion of the second overwintering stage of semivoltine species in long-term studies will be challenging but is urgently needed to shed light on how the shifting snow cover regimen can alter mass and overwintering metabolism on such species (see Konvička et al. [13]). Another important aspect for larval development and survival rate of the immature stages of these stenotopic butterflies would be the inclusion of specific microclimatic conditions [63].

## 5. Conclusions

We provided evidence of enhanced performance of *Erebia* caterpillars in their first developmental stage (L1, L2) in warmer conditions and a decreasing performance in colder conditions. Furthermore, we demonstrated that small movements from mother’s origin could be beneficial for species performance, but not large movements. Apparently, species do not adapt equally well to any temperature change. For example, warming did not benefit *E. nivalis* (at least at its first life stage), as biomass was substantially reduced in the warmest conditions, and cold reduced *E. aethiops* performance. Previous studies have demonstrated that mountain species inhabiting lower elevation areas also appear to have increased cold hardiness [28,64]. Our findings similarly suggest that wider elevational distribution may increase the ability of individuals to regulate their performance under various thermal conditions. In light of more frequent extreme weather events, ecologists have begun to consider this process [1,28,62]. Ideally, studies on multiple species with different traits and at different life history stages will help us to move beyond simple assumptions about performance of cold-adapted species (e.g., thermal tolerance increases with elevation) and improve future projections of climate change on alpine fauna. For example, Konvička et al. [13] found that climate warming among different life history stages creates a dynamic equilibrium that preserves mountain insect populations or even allows them to increase through time [13]. It is, therefore, imperative to gain a more mechanistic understanding of species’ thermal tolerance under extreme weather events because asynchronous changes during development will affect organismal survival, species persistence, community composition, and ecosystem processes and services.

## Figures and Tables

**Figure 1 biology-11-00915-f001:**
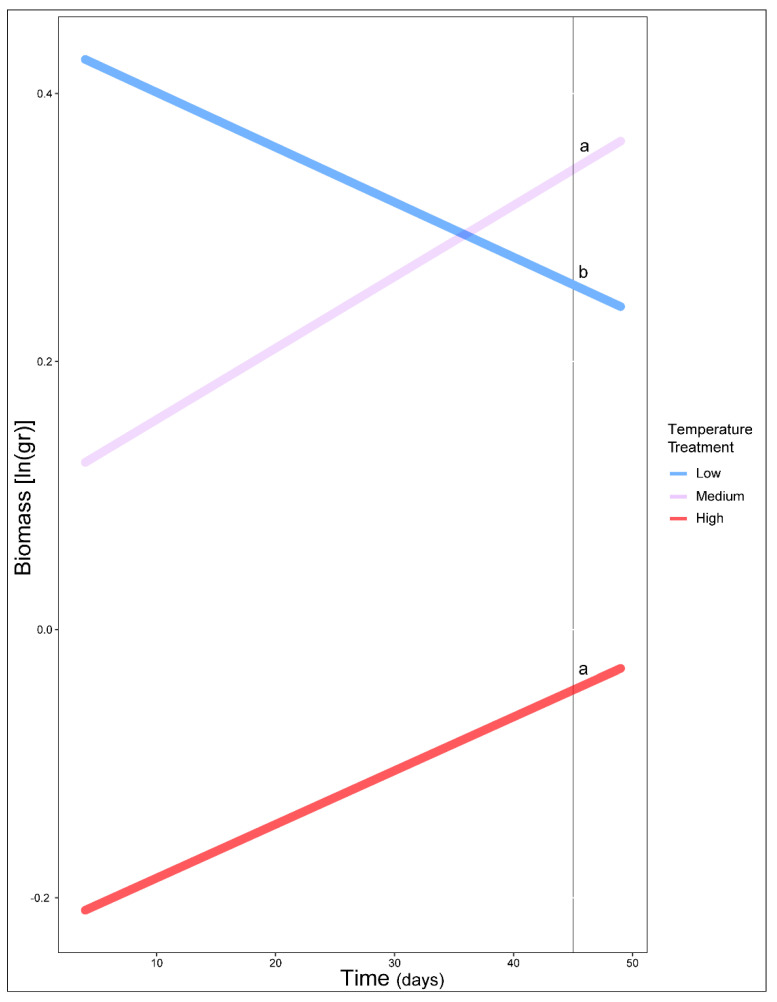
Estimated performance of *Erebia* caterpillars as predicted for the three temperature treatments (low, medium, high) and across time. Lowercase letters indicate differences between treatments. Model fitted for individuals that had two or more mass measurements. Note that hatching time slightly varied among larvae, resulting in different biomass at the beginning of weekly measurements (see Methods).

**Figure 2 biology-11-00915-f002:**
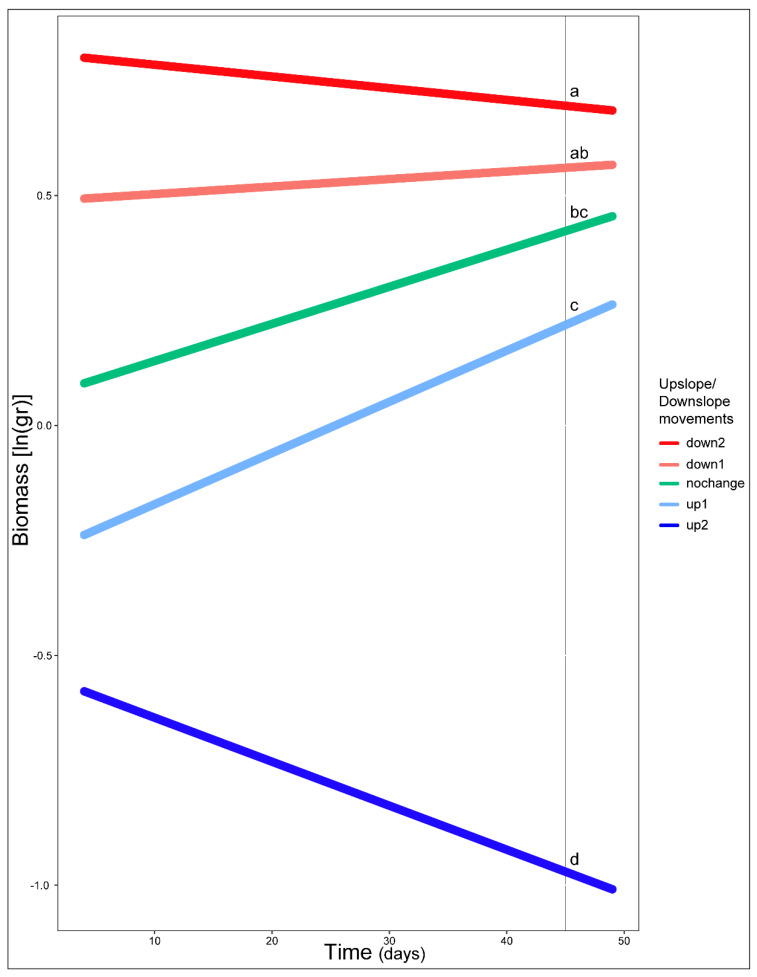
Estimated performance of *Erebia* caterpillars as predicted for upslope/downslope movements from mother’s origin or no movement at all across time. Lowercase letters indicate differences between treatments. Model fitted for individuals that had two or more mass measurements. Note that hatching time varied among larvae, resulting in different biomass at the beginning of weekly measurements (see Methods).

**Figure 3 biology-11-00915-f003:**
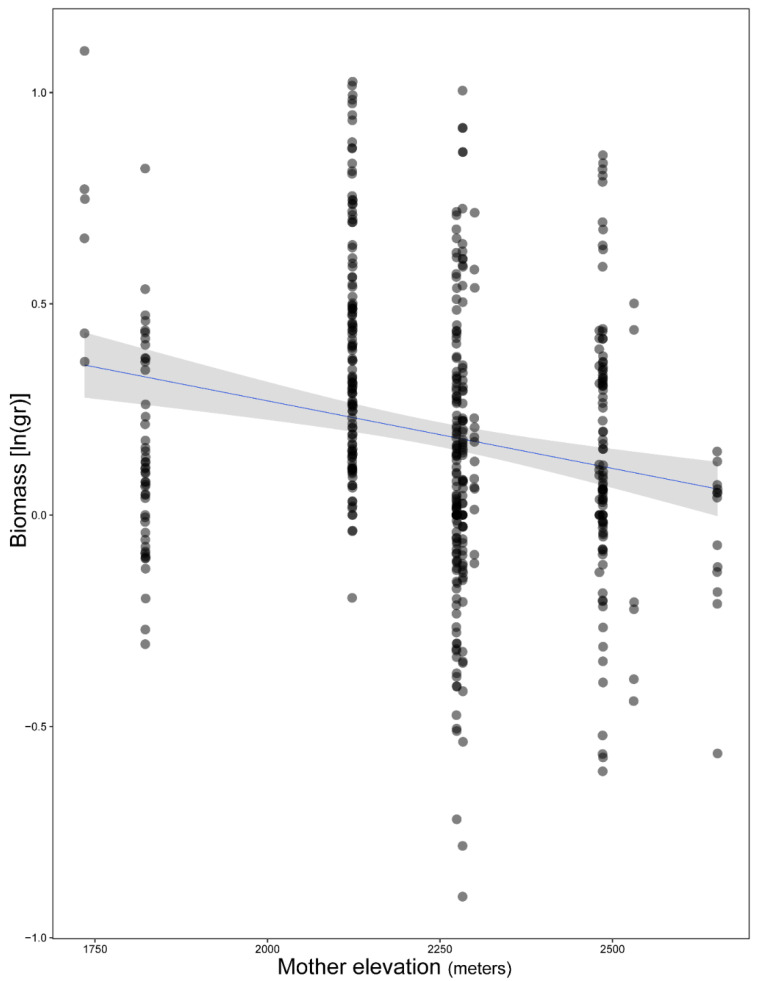
Relationship between *Erebia* caterpillars’ performance and mother’s elevation (i.e., sampling site elevation of mother’s origin). Each dot corresponds to an individual’s mass at a specific elevation.

**Figure 4 biology-11-00915-f004:**
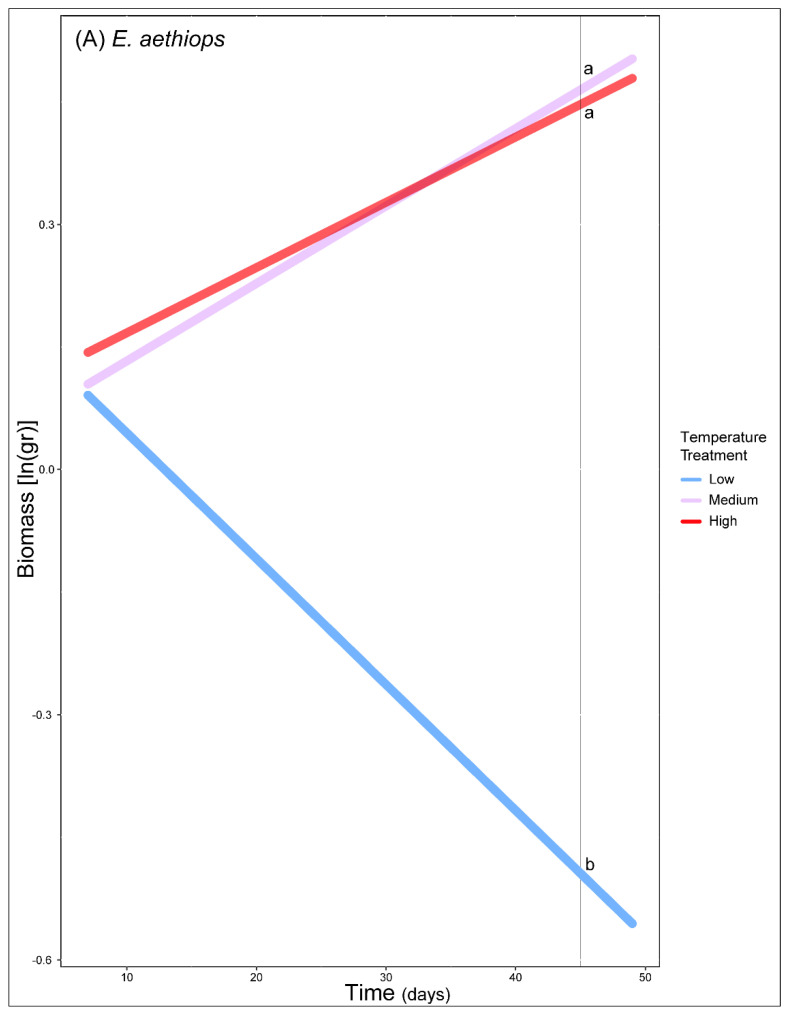
Estimated performance across time for (**A**) *E. aethiops* and (**B**) *E. cassioides*. Predicted mean values (center point) of biomass and the CI at 95% for the different treatments are plotted for (**C**) *E. tyndarus* and (**D**) *E. nivalis*. The different treatments are noted with lowercase letters. Models fitted for individuals that had two or more mass measurements. As we found no difference in the performance of *E. manto* among temperature treatments, the species is not illustrated.

## Data Availability

Data on caterpillars’ weight are all presented in Appendix A.

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
