# Peer review of "Not Too Warm, Not Too Cold: Thermal Treatments to Slightly Warmer or Colder Conditions from Mother’s Origin Can Enhance Performance of Montane Butterfly Larvae"

_biology, 2022, doi:10.3390/biology11060915_

Round 1

Reviewer 1 Report

I gather (because the presentation is not clear, as it should be) that the larvae were held at CONSTANT temperatures in all treatments, despite a fixed photoperiod. Perhaps larvae under snow experience a nearly constant T for extended periods; otherwise there is almost always a diel T rhythm. We know from both our own work and that of others that response to constant T (say, the mean of diel max and min) is usually NOT equivalent to a diel T cycle. (It is difficult to get precision regulation in standard incubators/growth chambers. We use a day and a night T setting and astronomical timers to advance the photoperiod cycle daily to mimic natural seasonality.) These results then can only be considered suggestive.

The causation of 20year cycles in alpine butterflies remains contentious. It is particularly striking that most or all of the semivoltine species in a given massif are usually synchronized, which seemingly rules out resource competition as an evolutionary cause, but suggests (?) parasiyoid avoidance. A peripheral issue to be sure, but one raised in the MS in section 2. (It is striking that one section of the genus Oeneis  is found in coniferous forest at relatively low elevations in both Eurasia and North America and does not "need" to have a 2-year cycle but does so anyway (phylogenetic inertia?). This might bear on the potential futures of such species in a warming world.

Author Response

Reviewer 1

We thank the reviewer for his/her critical and insightful comments that have greatly improved our manuscript. Please consider our feedback below.

I gather (because the presentation is not clear, as it should be) that the larvae were held at CONSTANT temperatures in all treatments, despite a fixed photoperiod. Perhaps larvae under snow experience a nearly constant T for extended periods; otherwise there is almost always a diel T rhythm. We know from both our own work and that of others that response to constant T (say, the mean of diel max and min) is usually NOT equivalent to a diel T cycle. (It is difficult to get precision regulation in standard incubators/growth chambers. We use a day and a night T setting and astronomical timers to advance the photoperiod cycle daily to mimic natural seasonality.) These results then can only be considered suggestive.

Reply: We thank the reviewer for this comment, and we agree that the presentation is not clear. The larvae were not held at constant temperatures, but temperatures changed with photoperiod between day and night. We used meteorological data available from the study area to mimic natural seasonality, and we adjusted values in the course of the experiment following seasonal astronomic daylength. For analytical purposes though, we considered the averaged value weighted by day and night length that was provided in Table S1.   
In the revised version, we improved the presentation of our methodology (please see section 2.2) in temperature treatments, while keeping in mind that someone might want to repeat our experimental design.

The causation of 20year cycles in alpine butterflies remains contentious. It is particularly striking that most or all of the semivoltine species in a given massif are usually synchronized, which seemingly rules out resource competition as an evolutionary cause, but suggests (?) parasiyoid avoidance. A peripheral issue to be sure, but one raised in the MS in section 2. (It is striking that one section of the genus Oeneis  is found in coniferous forest at relatively low elevations in both Eurasia and North America and does not "need" to have a 2-year cycle but does so anyway (phylogenetic inertia?). This might bear on the potential futures of such species in a warming world.

Reply: We adjusted our statement according to reviewer’s suggestions. Please see Lines 114-123.

Reviewer 2 Report

How the insects adapt the climate change biology is a hot topic for biologists. In this paper, the authors used five Erebia caterpillars species to investigate the performance under three thermal scenarios where conditions corresponded to the elevation of their mother’s origin. This work is very interesting and the data for the performance of these five species are solid. It is acceptable that  species’ performance in extreme high or low thermal treatments from that of mother’s origin can be harmed. Only several minor comments should be addressed before acceptance for publication. 1. Line 86. [5] = Vrba et al. [5]. 2. Line 120. (Sonderegger 2005) = [29]. 3. How to use the long term climatic data (1993-2019) to design the three temparature treatments? 4. References. The scientific name of the organisms should be italics. 5. What are the survival rates of the five used Erebia caterpillars species under three thermal scenarios?

Author Response

Reviewer 2

How the insects adapt the climate change biology is a hot topic for biologists. In this paper, the authors used five Erebia caterpillars species to investigate the performance under three thermal scenarios where conditions corresponded to the elevation of their mother’s origin. This work is very interesting and the data for the performance of these five species are solid. It is acceptable that  species’ performance in extreme high or low thermal treatments from that of mother’s origin can be harmed. Only several minor comments should be addressed before acceptance for publication.

We thank the reviewer for his/her positive spirit and careful reading of our MS. His/her crucial comments greatly improved our manuscript. Please consider our point-to-point feedback below. We feel that our joint work has resulted in a stronger manuscript.

1.Line 86. [5] = Vrba et al. [5].

Reply: Done

  1. Line 120. (Sonderegger 2005) = [29].

Reply: Done

  1. How to use the long term climatic data (1993-2019) to design the three temperature treatments?

Reply: We have enriched this part in our methodology. Please see Lines 126-133.

  1. References. The scientific name of the organisms should be italics.

Reply: Corrected.

  1. What are the survival rates of the five used Erebia caterpillars species under three thermal scenarios?

Reply: This is actually a very interesting point and it is one of our future goals to investigate potential variation on survival rate across different species and treatments. We would be happy to personally communicate to the reviewer any possible differentiation our future analysis might reveal.

Reviewer 3 Report

Biology MS 1725199 review report

This is an interesting and informative paper that warrants publication. The study is of interest and novel. The work has been performed well and the statistical approaches adopted are appropriate. The discussion is evidence based, but I would have thought that more could be said in relation to ‘what needs to be done next’. The manuscript does point to quite subtle, and often ignored, effects of changing thermal regimes in alpine areas and quite rightly point to the need for more mechanistic studies.  Two things that might be worth exploring in the future include longer studies on semivoltine species to include the second overwintering stage and linking the work to any ecological studies of overwintering site thermal regimes under changing snow cover and how this may interact with mass and overwintering metabolism.

There are a few minor comments, that can be easily addressed, as below:

Line 11. Is asynchrony just due to extreme weather condition? Perhaps better to say ‘Extreme weather events and climate change…’

Line 28 This does not fully reflect on the results (see lines 32-35). Given that the results describe the effects of the treatments on the individual species as well as overall, would it be better to state “Overall, we found…’’

Line 31 What are rigorous movements? Do you mean long distance movements? Perhaps replace with ‘ If individuals experience a large difference in thermal regime from  their maternal origin, organismal performance is harmed.’

Line 45 superscript “-18

Line 64 larvae are not eggs. ‘In Lepidoptera the immature stages (egg, larvae ad pupae) have limited or zero mobility…’

Line 96 This should be Erebia Dalman, 1816

Line 142 replace ‘visitation’ with ‘visit’

Line 146’.. that is distributed between…’ and again line 151

Line 148 delete ‘man-made’

Line 164 replace ‘into’ with ‘in’

Line 165 ‘The net served as barrier to contain the butterfly and was used as the main substrate for oviposition’

Line 195 ‘…, in normal potting…’

Line 198 superscript ‘2’

Line 200 -203 How frequently were the grass leaves changed?

Line 335  for clarity would it be better to say ‘the two semivoltine species’

Line 353 For clarity for this explanation, in the methods please state at what larval stage did the growth experiments stop, and here state that later larval stage performance was not monitored

Author Response

Reviewer 3

This is an interesting and informative paper that warrants publication. The study is of interest and novel. The work has been performed well and the statistical approaches adopted are appropriate. The discussion is evidence based, but I would have thought that more could be said in relation to ‘what needs to be done next’. The manuscript does point to quite subtle, and often ignored, effects of changing thermal regimes in alpine areas and quite rightly point to the need for more mechanistic studies. Two things that might be worth exploring in the future include longer studies on semivoltine species to include the second overwintering stage and linking the work to any ecological studies of overwintering site thermal regimes under changing snow cover and how this may interact with mass and overwintering metabolism.

There are a few minor comments, that can be easily addressed, as below:

We thank the reviewer for his/her positive spirit and helpful suggestions that greatly improved our manuscript. Following his/her suggestions we have rewritten misleading parts of MS to be easy for the readers. Please consider our point-to-point feedback below.

Line 11. Is asynchrony just due to extreme weather condition? Perhaps better to say ‘Extreme weather events and climate change…’

Reply: Done.

Line 28 This does not fully reflect on the results (see lines 32-35). Given that the results describe the effects of the treatments on the individual species as well as overall, would it be better to state “Overall, we found…’’

Reply: Done. 

Line 31 What are rigorous movements? Do you mean long distance movements? Perhaps replace with ‘ If individuals experience a large difference in thermal regime from  their maternal origin, organismal performance is harmed.’

Reply: We have rewritten this part for clarity. 

Line 45 superscript “-18

Reply: Done. 

Line 64 larvae are not eggs. ‘In Lepidoptera the immature stages (egg, larvae ad pupae) have limited or zero mobility…’

Reply: Done.

Line 96 This should be Erebia Dalman, 1816

Reply: Corrected.

Line 142 replace ‘visitation’ with ‘visit’

Reply: We’ve replaced it.

Line 146’.. that is distributed between…’ and again line 151

Reply: Corrected.

Line 148 delete ‘man-made’

Reply: Done.

Line 164 replace ‘into’ with ‘in’

Reply: Done.

Line 165 ‘The net served as barrier to contain the butterfly and was used as the main substrate for oviposition’

Reply: Sentence changed according to reviewer’s suggestion.

Line 195 ‘…, in normal potting…’

Reply: Corrected.

Line 198 superscript ‘2’

Reply: Corrected.

Line 200 -203 How frequently were the grass leaves changed?

Reply: Every other day, we added this information in the MS.

Line 335  for clarity would it be better to say ‘the two semivoltine species’

Reply: Corrected according to reviewer’s suggestion.

Line 353 For clarity for this explanation, in the methods please state at what larval stage did the growth experiments stop, and here state that later larval stage performance was not monitored

Reply:  We have clarified this point according to reviewer’s suggestions.

Round 2

Reviewer 1 Report

It would appear my objections have been met.

Author Response

We thank the reviewer for the time invested in our joint work. The MS has been edited by the English native speaker co-author of the MS for English grammar and wording.